# Evaluation of a Novel Tool for Screening Inadequate Food Intake in Age-Related Macular Degeneration Patients

**DOI:** 10.3390/nu11123031

**Published:** 2019-12-12

**Authors:** Diana Tang, Paul Mitchell, Gerald Liew, George Burlutsky, Victoria Flood, Bamini Gopinath

**Affiliations:** 1Centre for Vision Research, Department of Ophthalmology and Westmead Institute for Medical Research, University of Sydney, Sydney, NSW 2006, Australia; diana.tang@sydney.edu.au (D.T.); paul.mitchell@sydney.edu.au (P.M.); gerald.liew@sydney.edu.au (G.L.); george.burlutsky@sydney.edu.au (G.B.); 2Faculty of Health Sciences, University of Sydney, Sydney 2006, NSW 2151, Australia; vicki.flood@sydney.edu.au; 3Westmead Hospital, Western Sydney Local Health District, Westmead 2151, NSW 2151, Australia

**Keywords:** age-related macular degeneration, diet adequacy, diet screening tool, nutrition

## Abstract

Diet assessment tools provide valuable nutrition information in research and clinical settings. With growing evidence supporting dietary modification to delay development and progression of age-related macular degeneration (AMD), an AMD-specific diet assessment tool could encourage eye-care practitioners to refer patients in need of further dietary behavioural support to a dietitian and/or support network. Therefore, the aim of this study was to evaluate clinical use of a novel, short dietary questionnaire (SDQ-AMD) to screen for inadequate food intake in AMD patients by comparing it against a validated food frequency questionnaire (FFQ). Recruitment sources included Sydney-based private eye clinics and research databases (*N* = 155; 57% female; 78 ± 8 years). Scoring criteria based on the Australian Dietary Guidelines and dietary recommendations for AMD in literature were developed and applied to dietary data from the FFQ and SDQ-AMD. Bland–Altman plot of difference suggests agreement between the FFQ and SDQ-AMD as most mean difference scores were within the 95% CI (6.91, −9.94), and no significant bias between the scores as the mean score increased ((regression equation: y = 0.11x − 2.60) (95% CI: −0.058, 0.275, *p*-value = 0.20)). Scores were also significantly correlated (0.57, *p* ≤ 0.0001). The SDQ-AMD shows potential as a diet screening tool for clinical use, however, additional studies are warranted to validate the SDQ-AMD.

## 1. Introduction

Diet assessment methods provide valuable nutrition information in research and clinical settings. A number of tools are available for use, including recall surveys of usual intake (e.g., Food Frequency Questionnaire (FFQ)) or recent intake (e.g., 24 h recalls); weighed food records/diaries; and diet histories [1]. FFQs have been the preferred method in many large epidemiological studies for cost-effectiveness, option for self-administration, and representation of long-term usual intake [1]. One example of its use was in the Blue Mountains Eye Study (BMES), a landmark population cohort study investigating eye diseases in older Australians living in the Blue Mountains region. The study assessed usual dietary intake in the previous 12 months using a 145-item semiquantitative FFQ that had been validated against weighed food records [2,3,4]. In clinical practice however, a long FFQ may not be a viable tool due to the time-consuming and tedious nature of the questionnaire, especially amongst those with a vision impairment. Therefore, the development of a shorter tool such as a short recall tool that captures actual intake over a specified time period (e.g., 24 h, one week) may be more practical and appropriate in this setting [1].

In age-related macular degeneration (AMD), growing evidence supports that modifying risk factors such as poor diet can delay the development and progression of AMD [5]. Dietary recommendations for AMD also overlap with healthy eating practices that would benefit other common chronic conditions, including obesity, diabetes and cardiovascular diseases. However, the latest Australian National Health Survey (NHS) indicated that older adults continue to show poor diet adequacy, with 91.8% of adults over 65 years not meeting the dietary recommendations for fruit and vegetables [6]. This highlights a need for new approaches to encourage healthy diet practices in older Australians not only for the management of AMD but also for other chronic conditions. As eye-care practitioners are one of the first points of contact for a patient when it comes to AMD, research into developing a rapid and easy dietary tool for use by eye-care practitioners is warranted. A short dietary questionnaire (SDQ-AMD) that has been modified from a validated survey [7] could assist eye-care practitioners with screening patients’ overall diets for inadequate food intake and identifying those in need of dietary behavioural support via dietitian referral, where appropriate, and/or patient support networks such as the Macular Disease Foundation Australia (MDFA)—an Australian non-government organisation supporting those living with macular disease. As a multidisciplinary approach to healthcare has been widely encouraged for optimal patient outcomes, a similar approach is expected to benefit AMD patients [8]. The SDQ-AMD is currently used in a randomised controlled trial (RCT) to monitor dietary behaviours in AMD patients [9]. As this is a novel tool, reported food intake will be evaluated against the validated FFQ used in the BMES to assess comparability between recent actual intake and usual intake, respectively. Therefore, the aim of the current study is to evaluate the clinical use of the SDQ-AMD to screen for inadequate food intake by comparing it to the FFQ in AMD patients participating in this RCT.

## 2. Materials and Methods

### 2.1. Study Population

This study involved 155 participants who were enrolled in an RCT investigating the effectiveness of a telephone-delivered intervention to improve adherence to dietary recommendations for AMD management. The full protocol for the trial has been published previously [9] in Sydney, Australia. Recruitment occurred between June 2018 and July 2019, with participants recruited from Sydney-based private eye clinics, the University of Sydney and MDFA following online advertisements. Exclusion criteria were (a) lack of sufficient English fluency; (b) unwilling to participate in the 4 month intervention program; (c) inability to provide informed consent. The study was conducted in accordance to the Declaration of Helsinki, and the protocol for the RCT was approved by The University of Sydney Human Ethics Committee (Reference: HREC 2018/219).

### 2.2. Baseline Questionnaire

The self-administered baseline questionnaire was provided immediately after consent to participate was confirmed. It covered questions about general demographics, medical and surgical history, and visual function. Demographic questions collected information on ethnicity, marital status, living arrangements, level of education and employment status. Medical and surgical history questions provided an overview of the patient’s health status and any comorbidities, and other eye conditions like cataracts and glaucoma. Data was collected on AMD stage and lesions and the type and number of anti-VEGF injections patients received. The National Eye Institute Visual Function Questionnaire was also administered to assess general vision, difficulty with activities and other vision problems [10].

### 2.3. Food Frequency Questionnaire 

The 145-item self-administered FFQ was included along with the baseline questionnaire, with an estimated completion time of 30–40 min. Participants were asked to reflect on their usual intake in the previous 12 months of all major food groups, beverages and discretionary foods. It was adapted from the validated FFQ [2,11,12] that was used in BMES for older adults to include modern food trends such as nut milk and coconut oil. Frequency options included “Never”; “Less than 1 per month”; “1–3 per month”; “1 per week”; “2–4 per week”; “5–6 per week”; “1 per day”; “2–3 per day”; “4+ per day”. These were converted to fractions of one day to determine average daily intakes, that is, the above frequencies became “0”, “0.02”, “0.07”, “0.14”, “0.43”, “0.79”, “1”, “2.5”, and “4”. Predefined portion sizes for each FFQ item were listed. Qualitative questions were also included in the FFQ to capture types and brands of particular food items, with free-response questions to specify the cereals and oils consumed, and multiple-choice options for margarine (e.g., olive spread, reduced fat, cholesterol-lowering) and butter (e.g., ordinary, reduced fat, dairy blend). 

Preparation of FFQ data for statistical analyses involved converting frequency responses to grams consumed by multiplying the reported frequency and serve weight of the food item as listed in the FFQ or, if not specified, based on Australian Dietary Guidelines recommendations [13]. For free-response questions about cereals and oils used, the type and/or brands reported were manually matched to the Australian Food Composition Database [14] and linked to the frequency question to determine consumption values. Reported intakes were also adjusted to reflect seasonal variation based on Australian market data. 

### 2.4. Short Dietary Questionnaire (SDQ-AMD)

Following completion of the FFQ, the 36-item SDQ-AMD was administered by the study coordinator to reduce reporting errors and/or misinterpretation of questions. The survey had an estimated completion time of 10 min and was adapted from a validated questionnaire to capture intake in the last week of key food groups linked to AMD management. Some food items asked average number of serves consumed per day (fruits, vegetables, water) or per week (meat, seafood, legumes, nuts, eggs). The remainder of food items were asked as times consumed per week (e.g., bread, alcohol, fats and oils), under the assumption that each time consumed roughly equated to one serve [15].

### 2.5. Statistical Analysis 

Data from both FFQ and SDQ-AMD were entered into developed templates in REDCap and exported as Excel spreadsheets for statistical analysis in SAS version 9.4. 

To allow consistent comparison between the FFQ and SDQ-AMD, scoring criteria were developed. Scores and cut points were justified based on the Australian Dietary Guidelines [13] and existing literature on nutrition and AMD links (Online Appendix A) [5,7,13,16,17,18,19,20,21,22,23,24,25,26,27,28,29,30,31,32,33,34,35,36,37,38,39,40,41,42,43,44,45,46,47,48]. Food items included were total vegetables, dark green leafy vegetables, fruit, water, red meat, processed meat, fish/seafood, legumes, nuts, eggs, breads, cereals, select discretionary food and drinks, alcoholic beverages and fats and oil. To apply scores to FFQ data, the number of serves consumed for relevant food groups were calculated. For most food groups, calculations involved adding a participant’s total reported intake of food items within the same food group (e.g., adding up total grams of all listed fruits consumed) and dividing the total weight (in grams) by the standardised serve size as per the Australian Dietary Guidelines (e.g., 1 serve of any fresh fruit is approximately 150g) [13]. For food groups with food items of variable serve weights (e.g., specified discretionary food items), total energy (kJ) consumed of the food group was divided by the suggested serve size in kJ (e.g., 1 serve = 600 kJ). For alcohol, total alcohol intake in grams, rather than volume of alcohol consumed, was divided by one serve of alcohol (i.e., 10 g) [46]. For the SDQ-AMD, some additional data manipulation was required prior to applying score criteria, such as adjusting reported alcohol intake to reflect standard serves and converting reported weekly intake of specified discretionary items to daily intakes (Online Appendix A) [5,7,13,16,17,18,19,20,21,22,23,24,25,26,27,28,29,30,31,32,33,34,35,36,37,38,39,40,41,42,43,44,45,46,47,48]. For both FFQ and SDQ-AMD, appropriate scores from the scoring criteria were allocated based on serving intake, and total scores were recorded out of 30. Agreement and bias between the SDQ-AMD and FFQ were determined using the Bland–Altman point of difference, using total scores between the SDQ-AMD and FFQ against the mean difference in scores from both surveys. Limits of agreement (LOA) were ± 2SD from the mean difference. Pearson correlation coefficients were calculated for selected food groups from SDQ-AMD and FFQ based on daily serving intake and stratified by age (<80 years and ≥80 years). SDQ-AMD data that were reported as serves or times per week were divided by 7 to determine daily intakes. Correlation between FFQ and SDQ-AMD scores was also calculated. Correlation coefficients < 0.30 were considered negligible [49], and statistical significance was indicated by *p*-value < 0.05. Descriptive statistics were also used to report participant characteristics and dietary intakes. 

## 3. Results

Table 1 describes the study characteristics of the 155 participants. Most participants were Caucasian females (58%) and aged in their late 70s. Close to 95% of participants had a history of another medical condition, typically high blood pressure (63.2%) and arthritis (52.3%). Neovascular AMD was the most common type of AMD in this study (<80 years: 50%; ≥80 years: 61%), with cataracts also being a common eye condition affecting roughly 60% and 93% of participants <80 years and ≥80 years, respectively (Table 1). 

Figure 1 briefly describes the diets of the study participants. Less than half of AMD patients in this study met the suggested recommendations in the scoring criteria for nuts (47.7%), processed meat (38.1%), fruit (34.2%), eggs (31%), lower GI cereals (30%), overall vegetables (14.2%), legumes (12.9%), dark green leafy vegetables (6.5%) and discretionary foods or drinks (5.8%). The SDQ was administered 33.9 days (SD = 50.5 days) after FFQ completion, and Table 2 compares the average reported intakes from FFQ and SDQ-AMD. The strongest correlation coefficients across both age groups were observed for fish/seafood, nuts and alcohol (Table 2). Dietary scores between the SDQ-AMD and FFQ were also significantly correlated (0.57, *p* ≤ 0.0001). Average total scores were 13.6 (± 4.5) and 14.6 (± 4.1) out of 30 from the FFQ and SDQ-AMD, respectively. The Bland−Altman plot of difference illustrates that most mean difference scores lie within the 95% confidence interval and there was no significant bias between the scores as the mean score increased ((regression equation: y = 0.11 x − 2.60) (95% CI: −0.058, 0.275, *p*-value = 0.20)) (Figure 2).

## 4. Discussion

To the best of our knowledge, this is the first study to evaluate the use of the novel SDQ-AMD as a screening tool to identify inadequate food intake in AMD patients. Taking approximately 10 min to complete, compared with 30–40 min for the FFQ, this study suggests that the SDQ-AMD would be more appropriate in a clinical setting. 

Figure 1 shows the proportion of participants meeting the dietary recommendations. According to the latest NHS (2017–18) [6], which captured usual intake of fruit and vegetable serves per day with eight answer options ranging from “does not eat” to “6 serves or more”, overall adherence to fruit recommendations was higher in the general Australian population (65 years and over) than in the study participants (62.5% from NHS vs. 34.2% study participants), while consumption of recommended vegetable serves was comparably poor between groups (10.6% from NHS vs. 14.2% of study participants) [6]. Several factors contribute to inadequate dietary intakes in older adults, as well as those with AMD specifically. Individually, barriers including functional limitations (e.g., vision impairment), reduced appetite, limited dietary knowledge and cooking skills and low socioeconomic status [50] can limit the variety and amounts of food including fruits and vegetables consumed by older adults [51,52]. There are also limitations in clinical practice specific to AMD, including a lack of clear-cut guidelines on lifestyle and diet for patients and/or eye-care practitioners to follow; inadequate explanation and reinforcement by practitioners; and limited dietitian involvement and support [53,54,55].

The SDQ-AMD may help address some of these barriers as this study demonstrates that it could be appropriate for screening inadequate dietary intakes in AMD patients. Comparisons with the FFQ, as shown in Table 2, suggest that the SDQ-AMD has clinical relevance as the differences in reported average serve intakes between the two surveys were within 0.5 serve for most food groups with vegetables; included discretionary items, for example, biscuits and cakes; and water under-reported in the SDQ-AMD by approximately 1 serve on average. Association of reported intakes between surveys was determined using Pearson’s correlation coefficients, which ranged from 0.01 to 0.84. Just over half of the results showed negligible correlations (i.e., <0.30) [49]. On the other hand, a moderate positive correlation coefficient (0.57, *p*-value < 0.0001) was observed between survey total scores based on the scoring criteria (Online Appendix A) [5,7,13,16,17,18,19,20,21,22,23,24,25,26,27,28,29,30,31,32,33,34,35,36,37,38,39,40,41,42,43,44,45,46,47,48]. As correlation coefficients only indicate associations, a Bland–Altman plot of difference (Figure 2) was used to assess agreement between the SDQ-AMD and FFQ [56]. This test illustrates reasonably good agreement as most mean difference scores fell within the LOA. The line of regression also suggests that any bias between surveys is small. This is a promising finding as the FFQ has been adapted for Australian diets and validated for older Australians against weighed food diaries [2]. However, compared with other validation studies [7,57,58], a significant limitation of this study is a lack of direct comparison with weighed food diaries. As the SDQ-AMD and FFQ are both recall-based tools, inherent recall bias may not be accounted for in this study, and agreement between the two surveys could be overestimated [1]. Moreover, a second limitation of this study is the inconsistent administration of the SDQ-AMD, taking an average 33.9 days (SD = 50.5 days) after FFQ completion. A number of factors contributed to the varying timeframe, including participant availability, provision of incorrect contact details and lack of a specified timeframe for SDQ-AMD completion in the protocol for the RCT. This inconsistency could affect the quality of the data due to increased variability of external factors such as seasonal food habits. Unlike the FFQ, that captures seasonal food habits through the 12 month recall, the short recall period of the SDQ-AMD makes it more susceptible to seasonal dietary habits. Therefore, future studies should aim to compare the SDQ-AMD against multiple weighed food diaries to reliably validate the SDQ-AMD and administer these tests with minimal time lag. 

Despite the potential for recall bias and inconsistent administration of the SDQ-AMD, researchers attempted to improve data quality by minimising misinterpretation of questions when transitioning from the FFQ to SDQ-AMD by having the study coordinator administer the latter. This is because the self-administered FFQ consistently asked participants to reflect and report on their dietary intake in the previous 12 months, whereas the SDQ-AMD asked about intake in the last week, with questions changing from serves per day to serves per week to times per week. Although misinterpretation may also be a factor when completing the FFQ, it was successfully self-administered in the Blue Mountains Eye Study to report associations between diet and eye health, and time constraints and financial limitations did not allow for the study coordinator to administer it. 

Administration of the SDQ-AMD has also highlighted the need for modifications that might help to further improve the tool, such as (i) including a question to account for dairy food consumption to ensure a more comprehensive profile of habitual diets, especially with recent evidence [59,60] suggesting a protective effect of dairy foods on AMD risk; and (ii) converting the paper-based SDQ-AMD to an online format to make it readily accessible and facilitate automatic score calculations for instant feedback. Developing a simple, user-friendly version is essential to improve adherence, as a previous study utilising a mobile food diary application had a high drop-out rate due to high participant burden and design flaws (i.e., not user-friendly or application freezing) [61].

Therefore, a dietary tool like the SDQ-AMD has potential to assist practitioners with screening AMD patients for inadequate food intake and identifying those in need of further dietary behavioural support. As eye-care practitioners are the first to diagnose AMD, practitioners should encourage their patients to improve modifiable risk factors. According to a study on the clinical practice of Australian optometrists (*N* = 283), two-thirds reported providing dietary advice, and approximately 50% recommended nutritional supplements particularly for AMD cases, while less than half would discuss smoking [62]. Optometrists appear to be more open to providing diet and nutritional supplement advice to their patients compared with ophthalmologists, while ophthalmologists were more likely to educate their patients about the eye-health implications of smoking and encourage smoking cessation [63]. Consistent advice about modifiable risk factors by eye-care practitioners is encouraged. In the case for diet, use of the SDQ-AMD could help with this, and the dietary patterns of our study participants justify the need for intervention. If used by eye-care practitioners, the SDQ-AMD should serve to only prompt discussions aimed at raising patient awareness about the importance of dietary associations with AMD, encourage appropriate referral to dietitians for further thorough assessment and education and/or referral to advocacy groups such as the MDFA for general resources and information around diet and AMD links. 

We caution that our participants were predominantly receiving treatment for neovascular AMD, hence, these data are likely not representative of the general Australian population who are at risk of AMD development and/or have early stage AMD.

## 5. Conclusions

Study findings suggest that the SDQ-AMD has potential clinical use to screen for inadequate food intake in AMD patients. As a rapid and easy tool, it is more appropriate to administer in a clinical setting than the FFQ. Further studies are warranted to validate the SDQ-AMD, including comparison against weighed food diaries to account for recall bias. 

## Figures and Tables

**Figure 1 nutrients-11-03031-f001:**
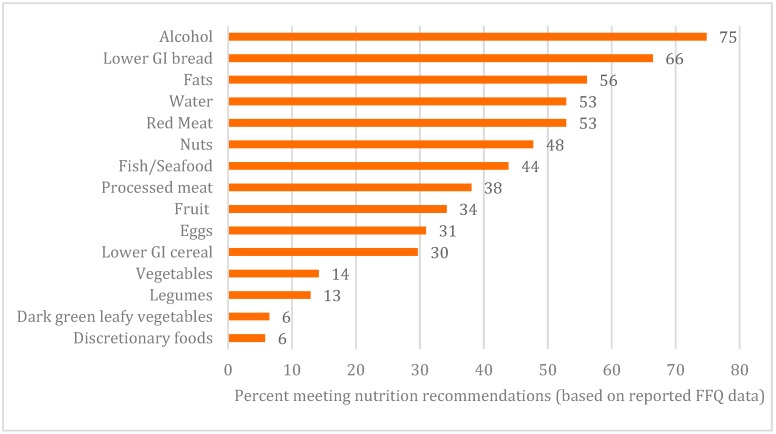
Proportion of participants meeting the recommendations for consumption of key food groups based on scoring criteria (Online Appendix A) [5,7,13,16,17,18,19,20,21,22,23,24,25,26,27,28,29,30,31,32,33,34,35,36,37,38,39,40,41,42,43,44,45,46,47,48].

**Figure 2 nutrients-11-03031-f002:**
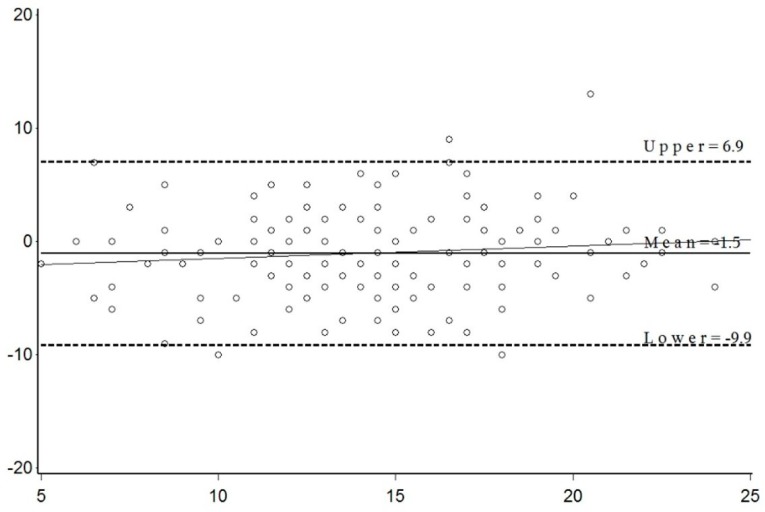
Bland–Altman plot of difference in score between the FFQ and SDQ-AMD versus the mean score of both methods. The solid line illustrates the mean difference (−1.5), and dotted lines represent the 95% confidence interval, with the upper and lower LOA (6.9 and −9.9, respectively) reflecting ± 2SD from the mean difference. Regression equation: y = 0.11 x − 2.60 (95% CI: −0.058, 0.275, *p*-value = 0.20). LOA: limits of agreement.

**Table 1 nutrients-11-03031-t001:** Participant characteristics stratified by age (<80 years, *n* = 86; ≥80 years, *n* = 69).

	<80 Years	≥80 Years
Age (years)	78 (5.7)	86 (3.5)
Females	53 (60.9)	36 (52.9)
Caucasian ethnicity	66 (79.5)	61 (93.8)
Lives alone (%)	13 (14.9)	23 (33.8)
History of other medical conditions (%):		
Heart attack (%)	7 (8.0)	10 (14.7)
Angina (%)	4 (4.6)	2 (2.9)
Other cardiac (%)	21 (24)	21 (30.9)
Stroke/Transient Ischemic Attack (%)	3 (3.4)	12 (17.6)
High blood pressure (%)	56 (64.4)	42 (61.8)
High cholesterol (%)	42 (48.3)	37 (54.4)
Diabetes/pre-diabetes (%)	21 (24.1)	15 (22.1)
Kidney disease (%)	6 (6.9)	4 (5.9)
Arthritis (%)	40 (46.0)	41 (60.3)
Other illness or major operation (%)	54 (62.1)	45 (66.2)
History of cataracts (%)	52 (59.8)	63 (92.6)
History of glaucoma (%)	9 (10.3)	14 (20.6)
Type of AMD by eyes	*N* = 140	*N* = 114
No AMD (%)	28 (20)	22 (19)
Early AMD (%)	6 (4.3)	0 (0.0)
Dry AMD (%)	24 (17)	8 (7.0)
Wet AMD (%)	70 (50)	69 (61)
Dry and Wet (%)	12 (8.6)	15 (13)
Eyes with wet AMD, receiving treatment with	*N* = 79	*N* = 80
Eylea (%)	54 (68)	52 (65)
Lucentis (%)	24 (30)	28 (35)
Avastin (%)	1 (1.3)	0 (0.0)

All data are presented as n (%) or mean (SD). AMD: age-related macular degeneration.

**Table 2 nutrients-11-03031-t002:** Comparison of the FFQ and SDQ-AMD in terms of average serves consumed per day, stratified by age (<80 years, *n* = 86; ≥80 years, *n* = 69).

Food Group	Age (years)	FFQ Daily Mean Intake (SD)	SDQ-AMD Daily Mean Intake (SD)	Correlation Coefficient *	*p*-Value **
Fruits	<80	1.79 (1.72)	1.84 (1.06)	0.19	0.08
	≥80	1.87 (2.80)	1.79 (1.07)	0.33	0.01
Vegetables	<80	3.20 (1.92)	2.25 (1.32)	0.15	0.18
	≥80	3.09 (2.07)	1.91 (1.10)	0.09	0.45
Dark green leafy	<80	0.11 (0.15)	0.16 (0.24)	0.18	0.09
vegetables	≥80	0.12 (0.17)	0.16 (0.29)	0.31	<0.01
Red meat	<80	0.37 (0.35)	0.28 (0.22)	0.16	0.13
	≥80	0.37 (0.27)	0.36 (0.23)	0.31	<0.01
Processed meat	<80	0.32 (0.35)	0.17 (0.23)	0.28	<0.01
	≥80	0.37 (0.40)	0.19 (0.23)	0.01	0.93
White Meat	<80	0.29 (0.35)	0.26 (0.18)	0.29	<0.01
	≥80	0.25 (0.22)	0.23 (0.19)	0.14	0.24
Fish/Seafood	<80	0.33 (0.28)	0.28 (0.29)	0.35	<0.001
	≥80	0.38 (0.35)	0.24 (0.17)	0.58	<0.0001
Eggs	<80	0.35 (0.25)	0.46 (0.33)	0.32	<0.01
	≥80	0.29 (0.28)	0.37 (0.27)	0.50	<0.0001
Legumes	<80	0.14 (0.24)	0.11 (0.17)	0.27	0.01
	≥80	0.11 (0.18)	0.10 (0.14)	0.07	0.57
Nuts	<80	0.65 (1.00)	0.55 (0.57)	0.48	<0.0001
	≥80	0.62 (1.02)	0.37 (0.49)	0.54	<0.0001
Low GI	<80	1.51 (1.50)	1.01 (0.86)	0.27	0.01
	≥80	1.34 (1.02)	1.16 (0.74)	0.50	<0.0001
High GI	<80	0.42 (0.54)	0.30 (0.49)	0.24	0.02
	≥80	0.56 (0.72)	0.36 (0.56)	0.36	<0.01
Biscuits and cakes, ice cream, sugary drinks, takeaway, processed potato	<80	2.01 (1.47)	1.10 (0.82)	0.43	<0.0001
	≥80	2.47 (1.63)	1.34 (1.18)	0.26	0.03
Water	<80	5.58 (2.19)	4.86 (2.58)	0.11	0.32
	≥80	5.71 (1.83)	4.31 (2.06)	0.46	<0.0001
Alcohol	<80	0.89 (1.69)	0.65 (1.40)	0.84	<0.0001
	≥80	0.69 (1.01)	0.47 (0.82)	0.81	<0.0001

* Correlation coefficients <0.30 considered negligible [49] ** Significance indicated by *p*-value < 0.05. FFQ: food frequency questionnaire. SDQ-AMD: short dietary questionnaire.

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
