# Peer review of "Evaluation of a Novel Tool for Screening Inadequate Food Intake in Age-Related Macular Degeneration Patients"

_nutrients, 2019, doi:10.3390/nu11123031_

Round 1

Reviewer 1 Report

Age-related macular degeneration is the leading cause of blindness in the elderly. It has been demonstrated that dietary intervention, as for instance modifying the diet by introducing foods reach in anti-oxidant and anti-inflammatory molecules, may delay the progression of the disease in patients suffering from this disease. In their paper, Tang and co-authors propose a short diet questionnaire adapted for AMD patients from a validated survey to screen patients for their diet in order to identify those at high risk, thus producing a valuable tool for eye-care practitioners. The paper is concise, well written and clearly state the limitations of the study that, as reported by the authors, cannot be considered representative of the general population that has not developed late AMD. I suggest performing a deep revision of the text for typos (for instance: the words “the trial.” should be removed in line 70; the word “requited” should be changed with “required” in line 120; the acronym LOA is introduced both in line 126 and in line 189).

Author Response

Thank you for your feedback. We appreciate the time you have spent reviewing this manuscript and have endeavoured to thoroughly revise the text for additional typos which has now improved the manuscript. 

Reviewer 2 Report

The number of subjects should be increased.

No new finding.

The conclusion is derived from a small number of subjects.

Many things can be involved in AMD development other than vegetables. In general eating, vegetables are good for overall health.

Author Response

Comment 1:  

The number of subjects should be increased. 

Response to Comment 1:  

Thank you for your feedback. We appreciate the time you have spent reviewing this manuscript. We agree that the number of subjects is small, and ideally would like to have had more participants involved. However, the 155 participants reported in this study are part of an ongoing randomised control trial, and for this trial a minimum of 140 participants was considered sufficient to achieve the primary outcome (i.e. 0.5 serve per day change in vegetable intake) with 80% power as significant at the 5% level, and allowing for 10% drop-out rate.   

Comment 2:  

No new finding. 

Response to Comment 2:  

Thank you for your comments. We acknowledge that the inadequate dietary intakes of older Australians as reported in our study is a common finding. However, we believe that our development, application and evaluation of the SDQ tool in AMD patients is a new finding that adds to the literature on nutrition and AMD. Moreover, the findings of our study have potential clinical implications, that is, the SDQ could be used as a screening tool by eye healthcare practitioners to screen AMD patients for poor dietary behaviours, which could allow for early implementation of targeted interventions aiming to improve dietary intakes in these patients. 

Comment 3:  

The conclusion is derived from a small number of subjects. 

Response to Comment 3:  

Thank you for your comments. As with comment 1, we agree that the number of subjects is small, and ideally would like to have had more participants involved. The 155 participants reported in this study, fulfils the minimum 140 participant requirement for the randomised control trial that study participants are currently involved in.  

Comment 4:  

Many things can be involved in AMD development other than vegetables. In general eating, vegetables are good for overall health. 

Response to Comment 4:  

Thank you for the final comments and the time you have spent reviewing the manuscript. We agree that many other factors are linked to AMD development, however with growing evidence to support that dietary intervention may delay the progression of AMD, we believe diet to be an important modifiable risk factor and therefore hope to identify inadequacies in the habitual diet of AMD patients using this new screening tool, SDQ-AMD, and encourage dietary intervention through dietitian referral and/or referral to support groups like the Macular Disease Foundation Australia which has nutrition information available on their website. 

Reviewer 3 Report

This study has been well written and adds to many other dietary studies that have been performed on AMD patients in other countries. There are some concerns however:

1. Within the introduction, there is no mention that a FFQ will be used to compare against the SDQ. It will be important to state this as currently, the reader is set up to think FFQs are a poor tool choice when used with patients living with visual impairment.

2. The introduction needs to have some stated aims. It is difficult to work out whether the point of the study was to compare the SDQ and the FFQ, or to investigate the diet of AMD patients, or both. The aims can then be addressed fully in the discussion.

3. The FFQ was used over 12 months, but the SDQ was used over a week. It is unclear how the comparison worked when the timescales are so different - was the data from the week concerned extracted from the FFQ? It currently reads as though the data from the FFQ was just scaled down to an 'average' week's intake, in which case the comparison does not really work.

4. There needs to be a discussion over the pros and cons of administered vs self-administered questionnaires, since the FFQ was self-administered and the SDQ was not.

Only one grammar error picked up: line 45 'overweight' - consider rephrasing.

Author Response

Comment 1:  

Within the introduction, there is no mention that a FFQ will be used to compare against the SDQ. It will be important to state this as currently, the reader is set up to think FFQs are a poor tool choice when used with patients living with visual impairment. 

Response to Comment 1:  

Thank you for your feedback. We have endeavoured to improve the introduction as suggested so that readers are aware about the comparison between the FFQ and SDQ-AMD tools (para 2, page 2). 

“As this is a novel tool, reported food intake will be evaluated against the validated FFQ used in the Blue Mountains Eye Study to assess comparability between recent actual intake and usual intake, respectively.” 

Comment 2(i): 

The introduction needs to have some stated aims. It is difficult to work out whether the point of the study was to compare the SDQ and the FFQ, or to investigate the diet of AMD patients, or both.  

Response to Comment 2 (i):  

Thank you for your suggestion. We apologise for the confusion regarding the study aims and we have now included a stated aim (para 2, page 2). We also note the comparison FFQ has been previously validated among a cohort of older people (Smith et al), and has been used extensively to report associations of diet and eye health.  

“Therefore, the aim of the current study is to evaluate the clinical use of the SDQ-AMD to screen for inadequate food intake by comparing it to the FFQ in AMD patients participating in this RCT.” 

Comment 2(ii): 

The aims can then be addressed fully in the discussion. 

Response to Comment 2 (ii): 

Thank you for your feedback. As mentioned in response to comment 1, we have clarified the study aim, and have accordingly endeavoured to improve the discussion around this. As substantial changes have been made please refer to the Discussion (starting page 6) for the changes.  

Comment 3 (i): 

The FFQ was used over 12 months, but the SDQ was used over a week. It is unclear how the comparison worked when the timescales are so different - was the data from the week concerned extracted from the FFQ?  

Response to Comment 3 (i): 

Thank you for your feedback. We apologise for the lack of clarity in our methodology and have strived to improve it accordingly. The FFQ was provided at baseline to participants who consented to participating in the trial. The FFQ was completed once, and asked participants to reflect on their usual intake in the previous 12 months rather than it being used over 12 months. Once completed and returned, the study coordinator administered the SDQ-AMD (33.9 days (SD=50.5 days) later) to capture actual intake in the last week. Therefore, data from the SDQ-AMD was not extracted from the FFQ but reflective of participants’ recalled intake the week before the study coordinator administered the SDQ-AMD. We also acknowledge that the reporting of dietary intake may vary between those reported as usual 12 month habits compared to the last week, especially as it took an average 33.9 days to administer the SDQ-AMD after FFQ completion, and have added this as a potential limitation in the Discussion. 

As seen under Food Frequency Questionnaire in the Materials and Methods (Para 1, page 3): “The 145-item self-administered FFQ was included along with the baseline questionnaire, with an estimated completion time of 30-40 minutes. Participants were asked to reflect on their usual intake in the previous 12 months.” 

As seen under Short Dietary Questionnaire (SDQ-AMD) in the Materials and Methods (Para 3, page 3): “Following completion of the FFQ, the 36-item SDQ-AMD was administered by the study coordinator 

As seen under Discussion (para 1, page 7); “a second limitation of this study is the inconsistent administration of the SDQ-AMD, taking an average 33.9 days (SD=50.5 days) after FFQ completion. A number of factors contributed to the varying timeframe including participant availability, provision of incorrect contact details, and lack of a specified timeframe for questionnaire completion in the protocol for the RCT. This inconsistency could affect the quality of the data due to increased variability of external factors such as seasonal food habits.” 

Comment 3 (ii):  

It currently reads as though the data from the FFQ was just scaled down to an 'average' week's intake, in which case the comparison does not really work. 

Response to Comment 3 (ii):  

We apologise again for the confusion regarding the methodology. As with our response for (i), we have reported the usual dietary intake of the previous 12 months according to responses from the FFQ, and recalled dietary intake in the last week from the SDQ-AMD. We acknowledge the differences in dietary intake observed may be related to the varying recall timeframes asked of the participants. Where the 12-month recall for the FFQ would cover seasonal elements of food habits, the SDQ-AMD would involve varying levels of seasonal food habits, especially as the SDQ-AMD was administered on average 33.9 days (SD = 50.5days) after FFQ completion. We have endeavoured to reflect comparable weekly data, by converting reported frequencies (e.g. 1 per week, 2-3 per week) to fractions of one day to determine average daily intakes for foods commonly eaten daily (e.g. fruit, vegetables), and weekly intakes for other foods e.g. red meat, fish, eggs. Comparison between a FFQ and short recall tool have occurred in previous literature (Gnagnarella P. et al 2018), but it nevertheless is a potential limitation in the comparison between the two recall surveys. 

Comment 4: 

There needs to be a discussion over the pros and cons of administered vs self-administered questionnaires, since the FFQ was self-administered and the SDQ was not. 

Response to Comment 4:  

Thank you for your feedback. We appreciate the advice you have provided and have endeavoured to improve our discussion as suggested to highlight the pros and cons of administered vs self-administered and why there was a difference in administration for the FFQ and SDQ-AMD (Para 2, page 7). 

“...researchers attempted to improve data quality by minimising misinterpretation of questions when transitioning from the FFQ to SDQ-AMD by having the study coordinator administer the latter. This is because the self-administered FFQ consistently asked participants to reflect and report on their dietary intake in the previous 12 months, whereas the SDQ-AMD asked about intake in the last week with questions changing from serves per day to serves per week to times per week. Although misinterpretation may also be a factor when completing the FFQ, it was successfully self-administered in the Blue Mountains Eye Study... and time constraints and financial limitations did not allow for the study coordinator to administer it.” 

Comment 5.  

Only one grammar error picked up: line 45 'overweight' - consider rephrasing. 

Response to Comment 5.  

Thank you for identifying this error. We have decided to remove ‘overweight’ from the text so that line 45 now reads as ‘including obesity, diabetes, and cardiovascular diseases.’ 

Reviewer 4 Report

The effect of diet on occurrence and progression of AMD is an important clinical topic. 

I believe there is an error in Figure 1 as a number of bars had no Y axis label.  In addition, although the authors refer to statistically significant, sub-optimal intakes of fruits and vegetables in their study, no statistics are shown for the items in Figure 1, or elsewhere in the manuscript.  These statistics should be included.

There is a highlight error in line #83.

Author Response

Comment 1:

I believe there is an error in Figure 1 as a number of bars had no Y axis label. 

Response to Comment 1:

Thank you for your feedback and identifying this editing error with Figure 1. It appears that as the figure was resized during formatting, the labels for a number of bars disappeared. We have resized the figure now to ensure all labels are visible.

Comment 2:

In addition, although the authors refer to statistically significant, sub-optimal intakes of fruits and vegetables in their study, no statistics are shown for the items in Figure 1, or elsewhere in the manuscript.  These statistics should be included.

Response to Comment 2:

Thank you for your feedback. We apologise for the lack of clarity of our results and our incorrect reference to statistically significant sub-optimal intakes of fruit and vegetables. We have endeavoured to improve the paper to avoid confusion. Upon review, we have decided to focus on the evaluation of the SDQ-AMD with the FFQ for this paper, and have only briefly described the baseline diets as shown in Figure 1. We felt that the statistical analyses on dietary intakes will have a greater impact in a future study that will include follow up data from the participants as part of the RCT, to measure change in intakes of fruit, vegetables as well as other food groups to determine if there has been any significant changes. 

The abstract which previously included results on the inadequate fruit and vegetable intake, now reads as "Diet assessment tools provide valuable nutrition information in research and clinical settings. With growing evidence supporting dietary modification to delay development and progression of age-related macular degeneration (AMD), an AMD-specific diet assessment tool could encourage eye-care practitioners to refer patients in need of further dietary behavioural support to a dietitian and/or support network. Therefore the aim of this study was to evaluate clinical use of a novel short dietary questionnaire (SDQ-AMD) to screen for inadequate food intake in AMD patients by comparing it against a validated food frequency questionnaire (FFQ). Recruitment sources included Sydney-based private eye clinics, and research databases (N=155; 57% female; 78±8 years). Scoring criteria based on the Australian Dietary Guidelines and dietary recommendations for AMD in literature were developed and applied to dietary data from the FFQ and SDQ-AMD. Bland-Altman plot of difference suggests agreement between the FFQ and SDQ-AMD as most mean difference scores were within the 95% CI (6.91, -9.94), and no significant bias between the scores as the mean score increased ((regression equation: y = 0.11x - 2.60) (95%CI: -0.058, 0.275, p-value = 0.20)). Scores were also significantly correlated (0.57, p-<0.0001). The SDQ-AMD shows potential as a diet screening tool for clinical use however, additional studies are warranted to validate the SDQ-AMD."

The Results (para 3, page 4) now read as "Figure 1 briefly describes the diets of the study participants. Less than half of AMD patients in this study met the suggested recommendations in the scoring criteria for ..."

Comment 3:

There is a highlight error in line #83.

Response to Comment 3:

Thank you for your feedback, and the time you have spent reviewing this manuscript. The highlight error in line #83 has now been deleted. 

Round 2

Reviewer 2 Report

Manuscript has improved. Authors have taken care of all the comments.